# Application of Multiple Linear Regression and Artificial Neural Networks for the Prediction of the Packing and Capsule Filling Performance of Coated and Plain Pellets Differing in Density and Size

**DOI:** 10.3390/pharmaceutics12030244

**Published:** 2020-03-08

**Authors:** Panagiotis Barmpalexis, Ioannis Partheniadis, Konstantina-Sepfora Mitra, Miltiadis Toskas, Labrini Papadopoulou, Ioannis Nikolakakis

**Affiliations:** 1Department of Pharmaceutical Technology, School of Pharmacy, Faculty of Health Sciences, Aristotle University of Thessaloniki, 54124 Thessaloniki, Greece; pbarmp@pharm.auth.gr (P.B.); ioanpart@pharm.auth.gr (I.P.); konstamm@pharm.auth.gr (K.-S.M.); toskasmi@pharm.auth.gr (M.T.); 2Department of Mineralogy-Petrology-Economic Geology, School of Geology, Faculty of Sciences, Aristotle University of Thessaloniki, 54124 Thessaloniki, Greece; lambrini@geo.auth.gr

**Keywords:** capsule filling, packing indices, pellet density, pellet size, coating, artificial neural networks, regression

## Abstract

Plain or coated pellets of different densities 1.45, 2.53, and 3.61 g/cc in two size ranges, small (380–550 μm) and large (700–1200 μm) (stereoscope/image analysis), were prepared according to experimental design using extrusion/spheronization. Multiple linear regression (MLR) and artificial neural networks (ANNs) were used to predict packing indices and capsule filling performance from the “apparent” pellet density (helium pycnometry). The dynamic packing of the pellets in tapped volumetric glass cylinders was evaluated using Kawakita’s parameter *a* and the angle of internal flow *θ*. The capsule filling was evaluated as maximum fill weight (*CFW*) and fill weight variation (*FWV*) using a semi-automatic machine that simulated filling with vibrating plate systems. The pellet density influenced the packing parameters *a* and *θ* as the main effect and the *CFW* and *FWV* as statistical interactions with the coating. The pellet size and coating also displayed interacting effects on *CFW,*
*FWV,* and *θ*. After coating, both small and large pellets behaved the same, demonstrating smooth filling and a low fill weight variation. Furthermore, none of the packing indices could predict the fill weight variation for the studied pellets, suggesting that the filling and packing of capsules with free-flowing pellets is influenced by details that were not accounted for in the tapping experiments. A prediction could be made by the application of MLR and ANNs. The former gave good predictions for the bulk/tap densities, *θ, CFW,* and *FWV* (R-squared of experimental vs. theoretical data >0.951). A comparison of the fitting models showed that a feed-forward backpropagation ANN model with six hidden units was superior to MLR in generalizing ability and prediction accuracy. The simplification of the ANN via magnitude-based pruning (MBP) and optimal brain damage (OBD), showed good data fitting, and therefore the derived ANN model can be simplified while maintaining predictability. These findings emphasize the importance of pellet density in the overall capsule filling process and the necessity to implement MLR/ANN into the development of pellet capsule filling operations.

## 1. Introduction

Multiple-unit dosage forms (pellets) offer both technological (spherical shape, narrow particle size distribution, easier application of coating) and therapeutic advantages (lower gastric time variation, lower risk of dumping, feasibility of combination therapy with pellets containing different drugs, or the same drug but different functional excipients for controlled-release). However, to administer the pellets they have to be filled into hard gelatin capsules. The filling methods employed are mostly based on gravitational feeding where the capsule shell forms the volumetric measure, and hence the success depends on the flow and packing ability of the pellets, which to a large extent is controlled by the micromeritic characteristics and surface treatment [1,2]. Although several studies have been carried out on the effects of particle size and shape, there is no information in the literature on the effect of pellet density on capsule filling. 

One of the earliest attempts to predict packing and capsule filling performance from the properties of individual components was made by Newton and Bader (1981) who developed a relationship between capsule fill weight and theoretical maximum bulk density [3]. Furthermore, a direct relationship between the angle of internal flow [3] and fill weight variation was established by Varthalis and Pilpel, and by Podczeck et al. [4,5]. However, modern trends in the pharmaceutical industry and the introduction of process analytical technology (PAT) necessitates new methodologies that can accurately predict the filling performance directly from the properties of the feed particulates. To improve the prediction accuracy [2] applied computer simulation based on a Monte Carlo technique and investigated the influence of pellet size, dispersity, shape, and aggregation on the filling of hard-shell capsules. The results were in general agreement with the experimental observations and also confirmed that above an aspect ratio of 1.2 filling reproducibility is reduced. Ali et al. [6] also used computer simulation to investigate the role of pellet size, shape, and filling method on the fill weight variability of encapsulated pellets by simulating pellet size and shape distributions. The variability was predicted for a variety of pellet sizes and shapes. Other newer works under investigation report feasibility of terahertz reflection measurements to predict relative densities of packed powders and capsule fill weight [7].

Design of experiments (DoE), supplemented with polynomial model fitting via multiple linear regression (MLR), is gaining acceptance as a prediction tool in pharmaceutical formulation work due to its simplicity, software availability, and the physical interpretation of the effects and interactions [8,9]. However, there are cases where high precision levels in conjunction with generalizing ability are required and MLR may not adequately satisfy these requirements [10]. In such cases, more sophisticated regression techniques, e.g., feed-forward artificial neural networks (ANNs), have been suggested [11]. They are biologically inspired, highly efficient machine learning regression techniques that act as universal function approximators for modelling high complex non-linear relationships [12]. Among the several advantages of ANN regression, the development of a single universal fitting model which is able to generalize several stages of solids formulation development is very desirable (i.e., to predict intermediate particulate behavior in early stages and product characteristics in late stages). 

In this work, MLR was compared with ANNs for predicting the packing characteristics and capsule filling performance of plain or coated pellets, with small or large sizes. Since pellets may contain ingredients of low density, e.g., organic drugs, or high density, e.g., inorganic excipients, density is an important factor (besides pellet size) to consider due to its influence on the packing and capsule fill weight [3]. Pellet shape is also important, but as long as the aspect ratio is kept below 1.2 its effect is controllable [1,2]. Pellets of different density were obtained using paracetamol (density 1.343 g/cc) as a low density modifying ingredient, calcium phosphate dehydrate (density 2.893 g/cc) as intermediate, and barium sulfate (density 4.675 g/cc) as a high density ingredient. These modifiers cover cases from low density pellets containing high drug content to high density pellets containing inorganic excipients such as diluents (e.g., calcium phosphate hydrate) or disintegrants of release modifiers (e.g., barium sulfate). To our knowledge, this is the first time that the effect of density on capsule filling has been examined and included in MLR and ANN predictive models of capsule filling performance.

## 2. Materials and Methods

### 2.1. Materials

The density modifiers were: paracetamol (PCT, Eu.Ph., ρ_s_ = 1.343 g/cc) gifted from Boehringer Ingelheim, Germany (via Boehringer Ingelheim Hellas, Greece); calcium hydrogen phosphate (CPH, Emcompress, ρ_s_ = 2.893 g/cc) from Edward Mendell New York, USA; barium sulfate (BSF, ρ_s_ = 4.675 g/cc) from VWR Cemicals, Monroeville, PA, USA. Microcrystalline cellulose (MCC, Avicel PH-101, lot 6950C) was obtained from FMC (Cork, Ireland) and was added in different proportions with the density modifiers. Polyvinylpyrrolidone (PVP K25, 21,000 g/mol) was obtained from BASF (Ludwigshafen, Germany) and was used as binder as 3% *w*/*v* solution in deionized water. 

### 2.2. Preparation of Pellets 

Pellets were prepared using extrusion/spheronization. About 20-g MCC/modifier powder mixtures were placed in jars which were mounted on a Turbula mixer (Type T2C, Willy Bachofen AG, Basel Switzerland) and tumbled for 15 min at 45 rpm. They were then transferred into a 500 mL capacity bowl fitted with a 3-blade paddle for wet mixing using PVP (3% *w*/*w* in deionized water) as a binder. The produced wet mass was extruded in a radial extruder (Model 20, Caleva, Dorset, UK), operated at 25 rpm. It was fitted with replaceable 0.5 mm or 1 mm aperture extrusion screens for the production of small or large pellets, respectively. The consumed quantities of binder/gram solids for the MCC/PCT mixtures were between 4.8 and 17.6 mL, for the MCC/CPH between 7.2 and 18.4 mL, and for the MCC/BSF between 5.4 and 18.5 mL. The wet mass was immediately transferred into a spheronizer (Model 120, Caleva) fitted with a cross-hatch plate and rotated for 10 min, with the small pellets at 940 rpm corresponding to a linear velocity of 5.92 m/s and the large pellets at 1440 rpm corresponding to a linear velocity of 9.07 m/s [9]. MCC/modifier pellets with different proportions, 10% to 80%, were prepared and different density ranges were obtained. Compositions were selected from each one that gave densities 1.45 g/cc, 2.53 g/cc, and 3.61 g/cc, respectively, according to the experimental design.

#### Coating Process

Some 10-g batches of dry pellets were suspended in a mini coater/drier (Caleva, Sturminster Newton, UK) and polymeric coating was applied by spraying with Opadry^®^ 200 (6% *w*/*v*) polyvinyl alcohol based aqueous film coating polymeric dispersion in water for 5 min at a rate of 2.1 mL/min. Talc was added in the dispersion as an anti-adherent at 0.5% *w*/*w* concentration after dispersing for 20 min at 10,000 rpm (Ultra-Turrax, IKA-Werke, Staufen, Germany). The inlet air temperature was 40 °C and after the application of the coating the pellets were kept under fluidization for 5 min to dry. The weight increase due to the coating was 2–2.5%.

### 2.3. Characterization of the Pellets

#### 2.3.1. Size, Shape, Moisture Content, and Density 

Pellet size and shape was determined using an image processing and analysis system comprised of a stereomicroscope, top cold light source (Olympus SZX9, Tokyo, Japan and Highlight 3100, Olympus Optical), video camera (VC-2512, Sanyo Electric, Osaka, Japan), and software (Quantimet 500, Cambridge, UK). About 100 pellets were examined at a total magnification of ×32.5. Mean pellet diameter was expressed as equivalent circle diameter (diameter of a circle with the same area as the projected pellet) and particle shape as aspect ratio (quotient of longest and shortest orthogonal dimension). Moisture content (% weight change on dry basis) of the pellets was determined using a moisture analyzer (Unibloc MOC63u; Shimadzu Corporation, Kyoto, Japan) (accuracy ±0.001 g) by exposing approximately 5 g samples at 105 °C for 30 min.

For the determination of pellet density, helium pycnometry was applied (Ultrapycnometer 1000, Quantachrome Instruments, Boynton Beach, Florida, FL, USA). The instrument was calibrated using a standard 7.0699 cm^3^ steel ball. Samples were accurately weighed (3 decimals) and purged for 10 min before measurement. Sample volume (average of 10 runs) was measured from the displaced gas. Measurements were made in triplicate and mean values and standard deviations were calculated. According to the USP 31, Chapter <699>, pycnometric density is a convenient measurement of the density of pharmaceutical powders. This differs from the granular density, where impenetrable voids or inaccessible pores may alter the measurement. In the case of pellets made by extrusion/spheronization, closed pores may form during hot air drying due to the operation of capillary forces, resulting in larger measured volume or lower density [13]. For this reason, the term “apparent” pellet density is adopted in this work. 

#### 2.3.2. SEM Microphotographs

Photomicrographs were taken with a scanning electron microscope (SEM) (JEOL JSM-6390LV, Tokyo, Japan) and the morphology of plain and coated pellets was examined. Photomicrographs of the cross sections of the pellets were also taken in backscattered electron mode in order to demonstrate the coating surface layer. Unfortunately, since the emission of backscattered electrons and/or brightness depends on the atomic number of the chemical elements in the pellet, the identification of the coating layer was only possible for pellets containing barium sulfate with a high atomic number.

### 2.4. Evaluation of the Packing Ability of Pellets

The packing state of particulates in columnar arrangements is expressed as bulk density (*p*_b_), defined by the quotient of the pellet mass and confining volume. It includes the contribution of inter-pellet voids but also depends on the pellet density. Its value increases after mechanical stressing such as tapping or vibration. It was determined after pouring a known weight of pellets into a 100 mL glass graduated cylinder and application of 300 taps (14 mm vertical drop, USP1, Erweka SVM 101, Heusenstamm, Germany). From the bulk (*p*_b_) and tap (*p*_t_) densities, the compressibility index (*CC*%) [14] equal to the density change relative to tap density was calculated. Since *CC*% only depends on two states of packing, its discriminative ability for the free-flowing pellets used in this study is low [15]. For this reason, the dynamic packing behavior of the pellets was evaluated from volumetric changes during tapping, using the Kawakita and Lude (1971) and Varthalis–Pilpel (1976) models [4,16]. The former of the two is expressed by Equation (1) where N is the number of taps, and *a* are constants related to maximum volume reduction and cohesiveness, respectively. *C* is the degree of volume reduction (Equation (2)).
N/C = N/*a* + 1/*ab*(1)
*C* = [(*V*_o_ − *V*_f_)/*V*_o_](2)

The Varthalis–Pilpel (V–P) model is expressed by Equation (3) where *ε* is the porosity of the powder given by Equation (4).
ε^2^/(1 − ε) = *Κ*_O_/*N*(3)
*ε* = 1 – (bulk density/“apparent” pellet density)(4)
*K*o in Equation (3) is the intercept of the ordinate found by plotting *K* = [*ε*^2^/(1−*ε*)] against *N*. The angle of internal flow (*θ*) is estimated from the slope of the straight-line plot of (*K*−*K*o) vs. *N*. 

### 2.5. Capsule Filling

Hard gelatin capsules were filled using a benchtop semi-automatic capsule-filling machine (ZUMA Milano, Italy). Filling was conducted by pouring the pellets into the capsule bodies resting in the holes of the vibrated plate. Capsule fill-weight (*CFW*) was determined from the weight of filled capsules after subtracting the mean weight of empty capsule shells. Twenty capsules were weighed (±0.1 mg) on an analytical balance (ADA 180, ADAM Equipment, Milton Keynes, UK), which was linked to a computer for transfer and data analysis (mean, SD). Fill weight variation (%*FWV*) was computed from the equation:%*FWV* = [(Σ(*x*_i_ − *x*^2^)/(*n*−1)]^0.5^ × 100/*x*(5)
where *x* is the net pellet weight of a capsule and *x* the mean weight of *n* = 20 capsules.

### 2.6. Design of Experiments (DoE)

Response surface methodology, following L-optimal design, and the quadratic mathematical model were employed. The factors were: “apparent” pellet density (X_1_, numeric) at three levels: 1.45, 2.53, and 3.61 g/c selected from preliminary trials on binary MCC/density modifier mixtures; mean pellet diameter (X_2_ categoric) at two levels, the low obtained using the small orifice (0.5 mm) screen extruder and the high using the large orifice (1 mm) screen; pellet coating (X_3_, categoric) at two levels, the low for plain (non-coated) and the high for coated. It consisted of 11 design points with 7 repetitions, making 18 runs in total. The response variables were: bulk density (Y_1_), tap density (Y_2_), Carr’s index (Y_3_), Kawakita parameter *a* (Y_4_), angle of internal flow (Y_5_), capsule fill weight (Y_6_), and fill weight variation (Y_7_). Experiments followed an L-optimal design and were conducted in a randomized order and in triplicate. The DoE are shown in Table 1.

Significant models and model terms were estimated by multiple linear regression (MLR) using p-value 0.05 as a criterion and backward elimination. Polynomial (6) and quadratic (7) equations were derived based on the results of statistical analysis ANOVA.
Y_i_ = b_0_ + Σ(b_i_X_i_) + Σ(b_ij_X_i_X_j_)(6)
Y_i_ = b_0_ + Σ(b_i_X_i_) + Σ(b_ij_X_i_X_j_) + Σ(b_ii_X_i_^2^)(7)
Y_i_ is the measured response; b_0_ is an intercept term; b_i_, b_ij_, b_ii_ are regression coefficients for the main effects, two-way interactions, and quadratic terms, respectively. X_i_ are coded levels of the experimental factors. Models with p-values <0.05 and with lowest predicted residual sum of squares (PRESS) were selected as best fitting for comparisons with ANNs. For the construction and analysis of the models, the Design-Expert software 12 (Stat-Ease Inc., MN, USA) was used.

### 2.7. Artificial Neural Networks (ANNs) 

Feed-forward back-propagation ANNs were constructed. The DoE data were split into training (70%) and validation (30%) subsets based on the Kennard–Stone design, or “uniform mapping algorithm” [17]. The number of training cycles was selected on the basis of the mean squared error of prediction (MSEp) for the validation subset, according to the “early stopping” method that prevents network overtraining, i.e., memorizing noise [11]. To identify the optimum architecture of ANNs, preliminary trial-and-error tests were employed in networks containing 2 to 12 hidden units, using the “vanilla” or standard back-propagation (StBack) training algorithm. The learning rate was set at 0.2 and the maximum tolerated difference between target and output values was set to 0. The logistic sigmoid activation function (Equation (8)) was selected for all units. All input and output data (i.e., factors and response variables) were scaled from 0 to 1.
(8)$$f\left(x\right)={\left[1+\exp\left(-x\right)\right]}^{-1}$$

After determining the optimum ANN structure, the network was trained with other two algorithms, the resilient back-propagation (Rprop) [18], and the backpropagation momentum (BackMom) [19], for comparison. In Rprop, the maximum update value and weight decay exponent were set at 30 and 5, respectively, and the momentum term in BackMom was set at 0.5. The network’s sensitivity (i.e., change in the output for a given change in the input variable) and saliency (i.e., increase in the error function when an input is omitted from the network), also known as causal and predictive importance, were estimated in order to evaluate the input factor’s significance [20]. Additionally, two pruning algorithms, namely magnitude-based pruning (MBP) and optimal brain damage (OBD), were utilized to simplify the optimum-trained network. In MBP, after each training the link with the smallest weight was removed, while in OBD the changes of the error function after pruning a certain weight was approximated using Taylor series (Stuttgart Neural Network Simulator, SNNS, User Manual, Version 4.2). The Java Neural Network Simulator (JavaNNS version 1.1 for WIN32) software package (new version of Stuttgart Neural Network Simulator (SNNS) 4.2 kernel) was employed for the development, training, validation, and pruning of ANNs (http://www.ra.cs.uni-tuebingen.de/ downloads/SNNS/).

#### External Validation of MLR and ANN Models

The generalizing ability (predictive performance) of the selected MLR models and the original or pruned ANNs was tested on a randomly selected external validation set shown in Table 2. Pellets were prepared on the basis of the selected test runs, and the packing and capsule filling parameters (responses Y_1_, Y_2_, … Y_7_) were determined. The coefficient of determination (*R*^2^) of the observed vs. the predicted response values was used to compare the performance of the used fitting models.

## 3. Results and Discussion

### 3.1. Preparation and Characteristics of the Pellets

In Table 1, the pellet codes, the composition of solids in the pellets, and the volume of liquid binder required per 20 g solids to form spherical pellets with the narrowest size distribution are shown together with the corresponding “apparent” densities and the characteristics of pellet size distributions. The presentation of the data follows the arrangement of the experimental design. Data for high density/large size/plain pellets (F19) are included to enable direct comparisons, although this point is not part of the design. The average diameter of the pellets prepared with the 0.5 mm orifice extruder screen was 0.475 mm, and the average mean diameter of those prepared with the 1.0 mm screen was 0.865 mm. These are less than the respective orifice diameters due to water loss or mass reduction during drying of the wet pellets. Besides MCC and density modifier, the final dry pellets contain a small amount of PVP binder which was added to improve pellet quality due to the high modifier content needed to achieve the desired density levels. Since the volume of the 3% *w*/*v* binder solution required for pelletization was between 7.2 and 8.3 mL, an average volume of 7.75 mL was used and the PVP in the final pellets was 0.232 g (= 0.03 × 7.75) or 1.15%. The moisture contents were low between 1.35% and 1.91% (Table 1). Therefore, the introduction of noise into the pycnometry data due to MC% differences is not expected to be significant.

In Figure 1, representative images of plain pellets of the three densifiers: paracetamol (MCC/PCT), calcium phosphate hydrate (MCC/CPH), and barium sulphate (MCC/BSF) are presented. They appear spherical with low aspect ratios (<1.2), and hence, the shape effect was controlled. Furthermore, in Figure 1, a cross section of a coated MCC/BSF pellet is presented, demonstrating a dark surface layer against a bright pellet interior. The brightness was due to the emission of backscattered electrons by the barium, which has a high atomic number, whereas the dark layer represents polymeric coating. Unfortunately, such a presentation was not possible for the pellets of the other two densifiers because they did not contain chemicals with high atomic numbers.

### 3.2. Selection of “Apparent” Density Levels

Pellets with different density modifiers in proportions from 10% to 80% with MCC were prepared. Changes of the “apparent” pellet density (*p*_app_) due to the addition of a modifier are shown in Figure 2a–c. The addition of the low density (PCT) modifier with a density lower than MCC decreased *p*_app_, whereas the addition of the medium (CPH) and high density (BSF) modifiers with greater density increased *p*_app_. The obtained *p*_app_ ranges were: 1.34–1.68 g/cc for MCC/PCT, 1.68–2.89 g/cc for MCC/CPH, and 1.68–4.68 g/cc for MCC/BSF. The two lines in each subfigure correspond to the experimental or “apparent” (solid symbols) and to the theoretical pellet densities, calculated from those of the primary components (for MCC 1.682 g/cc, for PCT 1.343, for CPH 2.893, and for BSF 4.675 g/cc) using Equation (9).
*p*_theoretical_ = (*p*_modified_ × %modifier) + (*p*_MCC_ × %MCC) + (*p*_PVP_ × %PVP)(9)

The %PVP was found from the mL of the 3% *w*/*w* solution required for pelletization (Table 1). From the experimental density curves of each MCC/modifier combination, compositions were selected that gave *p*_app_ 1.45 g/cc, 2.53 g/cc, and 3.61 g/cc, respectively, according to the experimental design.

From Figure 2, it can be seen that the experimental curves always lie below the theoretical. The deviations were greater when larger proportions of MCC were added, which was ascribed to the closed pores (inaccessible to Helium) that are formed during the drying of the wet pellets, resulting in a larger measured volume or lower density [13]. In the case of coated pellets (coating contained about 2% hydroxyl propyl methyl cellulose of density 1.33 g/cc), a small density decrease was expected: from 2.53 to 2.51 for the MCC/CPH, and from 3.61 to 3.57 for the MCC/BSF pellets (there was no significant change for MCC/PCT pellets). For this reason, in the implementation of the predictive models, average values of 2.52 and 3.59 g/cc were used in the last two cases. 

### 3.3. Dynamic Packing

In Figure 3 and Figure 4, Kawakita and V–P plots, respective of densification vs. tapping number, are presented for small/plain (a), large/plain (b), small/coated (c), and large/coated pellets (d), and in Table 3, the values of packing parameters are shown together with results of capsule filling. The lines in each subfigure of Figure 3 and Figure 4 represent the pellets of the same size and treatment but different density (*p*_app_) and were constructed according to Equations (1) and (3). It is evident that straight lines were obtained confirming linearity of the Kawakita and V–P models. The lines in the former plots are close to each other, making differentiation difficult. Conversely, in the V–P plots, their position is clearly different, indicating that this model differentiates better pellet packing in terms of *p*_app_. In all cases, the V–P plots followed the same trend, i.e., the slope or angle of internal flow (*θ*), increased, i.e., the packing ability decreased as *p*_app_ increased (black symbols lowest, blue highest *p*_app_). The decreased packing ability of the high *p*_app_ pellets was ascribed to their inability to efficiently absorb and transform the supplied mechanical energy from tapping into mobility and rearrangement within the pellet bed. 

Regarding the effects of pellet size and coating on the packing parameters, effects were found only for the intermediate density (MCC/PCT) pellets, but not for the low (MCC/PCT) or the high (MCC/BSF) density pellets. Comparing small with large pellets of intermediate *p*_app_ (red) in plain form (Figure 4a,b) it appears that the slope of the lines for the large pellets were smaller than for the small pellets, indicating a lower *θ*, or better packing ability. This was also deduced from the values of the packing indices for the intermediate *p*_app_ pellets in Table 3. Comparing small/plain (F10–F11) with large/plain (F12–F14) pellets, it was observed that both bulk (*p*_b_) and tap density (*p*_t_) increased from 0.76–0.78 to 0.82–0.84 and from 0.90–0.91 to 0.93–0.95, respectively, whereas the packing indices (*CC*%, *a*, *θ*) decreased (from 14.90–16.05 to 10.83–11.73, from 0.16 to 0.11–0.13, and from 48.66–49.38 to 45.42–46.56. Conversely, the comparison between small/coated and large/coated pellets (F6–F7 vs. F8–F9) did not show significant changes of packing parameters (for *p*_b_ from 0.83–0.85 to 0.80–0.83 for *p*_t_ from 0.93–0.95 to 0.90–0.93, for CC% from 12.24–12.47 to 11.0–11.8, for *a* from 0.13–0.14 to 0.12–0.13, and for *θ* from 45.92–46.74 to 46.67–48.86). Therefore, the coating eliminated differences due to pellet size. 

Next, we compared the effect of the coating again for the intermediate density pellets. From Figure 4a,d (the cascade position) it appears that the slope of the line corresponding to small/coated pellets (Figure 4d) is smaller than that of the small/plain pellets (Figure 4a), indicating better packing. This can also be observed from the results in Table 4 by comparing small/plain (F10–F11) with small/coated pellets (F6–F7). The values of both *p*_b_ and *p*_t_ increased (from 0.76–0.78 to 0.83–0.85 and from 0.90–0.91 to 0.93–0.95, respectively) whereas the packing indices *CC*%, *a*, and *θ* decreased (from 14.90–16.05 to 12.24–12.47, from 0.16 to 0.13–0.14, and from 48.66–49.38 to 45.92–46.74, respectively). Conversely, when large/plain (F13–F14) with large/coated pellets (F8–F9) were compared, the differences were seen to be small or negligible (for *p*_b_ from 0.82–0.84 to 0.80–0.83, for *p*_t_ from 0.93–0.95 to 0.90–0.93, for CC% from 10.83–11.73 to 11.0–11.8, for *a* from 0.11–0.13 to 0.12–0.13, and for *θ* from 45.42–46.56 to 46.67–48.86). There was no significant change in the packing parameters due to pellet size or coating for pellets of low or high density (F1–F5 and F15–F18). The above results show that the coating affected the packing of small pellets, but not large pellets. 

Turning to the effects of the studied factors on capsule fill weight (*CFW*) and fill weight variation (*FWV*), it can be observed from Table 3 that, as expected, density has a major influence on *CFW*. For the low *p*_app_ pellets the *CFW* range was 407.9–490.4 mg, for the intermediate pellets 596.9–685.8 mg, and for the high pellets 770.6–829.4 mg. The pellet size did not affect fill weight, whereas the influence of coating varied. For the low *p*_app_ pellets, it caused a decrease from 450.7–464.1 to 407.8 mg (F1) for the small pellets, and from 490.4 (F5) to 417.4 mg (F2) for the large. For the intermediate *p*_app_ pellets, it caused a small increase from 621.4–627 (F6–F7) to 636.8–658.7 mg (F10–F11) for the small pellets, but a decrease from 596.9–604.3 (F8–F9) to 660.5–685.8 mg (F12–F14) for the large. For the high *p*_app_, it caused an increase from 770.6 (F18) to 797.5 mg (F15) for the small pellets, but a decrease from 829.4 (F19) to 806.3 mg (F16) for the large. The influence of the studied factors on *FWV* was not immediately obvious, but it appeared that high density pellets (F15–F19) showed less variation than those of low (F1–F5) and intermediate *p*_app_ (F6–F14) (0.90–2.29% compared to 0.91–3.61% and 0.92–3.49%, respectively). Overall, the application of the coating decreased *FWV*. This can be observed by comparing the *FWV* range 0.90–2.1% of coated (F3–F5, F10–F14, F18, F19) with 1.73–3.61% of plain pellets. 

### 3.4. Correlations between Packing and Capsule Filling Parameters

In Figure 5a–c plots of *CFW* vs. bulk (*p*_b_) or tap (*p*_t_) density and *FWV* vs. Kawakita’s *a* and *CC*% are presented. A linear increase in *CFW* with *p*_b_ (circles) or *p*_t_ (squares) is notable in Figure 5a, which was expected from the results of earlier works [3,7]. Although the correlation coefficients were relatively high (*R*^2^ 0.975 and 0.982, respectively) the scatter of the data restricted an accurate prediction. The previous work also demonstrated correlations between packing indices and *FWV* for relatively free-flowing powders in the size range 64–430 μm (Figure 1 in the previous paper). Obviously, from Figure 5b–c it is clear that with the present experimental pellets there is no such correlation. A trend of the increase in *FWV* with the packing index was seen at higher index ranges values, for Kawakita’s *a* 0.13–0.16, and for CC% 12.24–16.05, which represents pellets classified in terms of flowability as “good” but not “excellent” [15]. Therefore, unlike powders, predictions of capsule fill weight variation from Kawakita’s *a* and Carr’s *CC*% indices that are described in the USP Chapter < 616 > for powders are not possible for spherical shape pellets with mean diameters greater than 0.475 mm. This finding suggests that the filling and packing of capsules with free-flowing pellets was influenced by details of the filling methods that were not accounted for in the tapping experiments. 

### 3.5. Multiple Linear Regression Analysis 

To overcome the inability of the packing indices to accurately predict the capsule filling performance, statistical multiple linear regression (MLR) combined with polynomial model fitting, and artificial neural networks (ANNs) regression were applied for the prediction of: bulk density (*p*_b_), tap density (*p*_t_), Carr’s index (*CC*%), Kawakita’s parameter (*a*), angle of internal friction (*θ*), capsule filling weight (*CFW*), and capsule fill weight variation (*FWV*) from the “apparent” density (X_1_) for different levels of the categorical factors pellet size (X_2_) and coating (X_3_). In Table 4, statistically significant polynomial models derived by application of MLR to the data of Table 3 are shown together with significant terms and goodness of fitting (R^2^). For each response variable, model equations are shown for the two levels of the categorical factors X_2_, X_3_ when their effect is included. Models for *θ*, *CFW*, and *FWV* were hierarchically corrected by adding a non-significant “parental” factor X_2_. Before entering the analysis, Y_3_ and Y_7_ responses were mathematically transformed (power of 1.98 and natural log transformation, respectively) in order to meet the assumptions of ANOVA for normal distribution of residuals with a constant variance. The equations in Table 4 are expressed in terms of X_1_ because this is the only numerical variable in the design. Variables X_2_, X_3_ are categorical, and where their effect was significant (e.g., for responses Y_3_–Y_7_), the values 0 or 1 are substituted into the equations for X_2_ or X_3_ as follows: for small size pellets X_2_ = 0, and for large size X_2_ = 1. For plain (non-coated) pellets X_3_ = 0, and for coated pellets X_3_ = 1. It was observed in Table 4 that the p-values of the intercepts were significant, i.e., for zero density (X_1_) or the absence of matter, packing indices and capsule filling parameters had values, which is nonsensical. This is because X_1_ = 0 is outside of the experimental range 1.457–3.619 g/cc of pellet density and has no meaningful interpretation. However, intercepts were included in the regression models to increase their predicting ability, which was the purpose of the study. 

From Table 4, it appears that for bulk density (Y_1_), tap density (Y_2_), angle of internal flow (Y_3_), capsule fill weight (Y_6_), and fill weight variation (Y_7_), the models provide good fitting with R^2^ > 0.964, but for *CC*% and the Kawakita’s *a,* the fitting is not as good (R^2^ 0.639 and 0.705, respectively). *p*_b_ is described by a single term (density, X_1_) linear model and *p*_t_ by a simple polynomial including density as linear (X_1_) and quadratic (X_1_^2^) term. *θ* and *FWV* are described by a polynomial, having all factors as linear (X_1_, X_2_, X_3_), density as quadratic (X_1_^2^), and an interaction term for the effects of pellet size and coating (X_1_X_3_). Lastly, *FWV* is described by a polynomial with all factors expressed by linear terms, besides a quadratic term for density, an interaction term for the effects of density and coating (X_1_X_3_), and another interaction term for the effects of pellet size and coating (X_2_X_3_). 

As a further step, the significant quadratic and interaction terms in the above models were graphically visualized in Figure 6. The quadratic effects of density on *p*_t_, Kawakita’s *a* and *θ* are evidenced in Figure 6a–c as curvatures of the initially straight lines at higher density values. The curves have decreasing slopes which are expressed by negative terms in the respective equations (Table 4, rows B, C, D). This signifies that the effect of *p*_app_ was more important at low to intermediate densities which is attributed to the inability of the high *p*_app_ pellets to absorb the mechanical energy from tapping and transform it into mobility for better rearrangement and packing. 

Interaction plots for the effects of pellet diameter and coating on *θ*, *CFW*, and *FWV* are shown in Figure 6d–f for the effects of *p*_app_ and coating, and in Figure 6g for the effects of *p*_app_ and coating on *CFW*. From Figure 6d–f it appears that after coating (dotted/red lines) there is no change in *θ*, *CFW*, and *FWV* regardless of pellet size, whereas for plain (non-coated) pellets (solid/green lines) there is a small decrease in *θ* and a small increase in *CFW* for the large pellets (Figure 6d,e, respectively) but a large decrease in *FWV* for the large pellets (Figure 6f). These interaction plots indicate that after coating, both small and large pellets behave the same, demonstrating smooth filling and low fill weight variation. The slight increase in *θ* and *CFW* might be ascribed to some surface sticking due to the presence of polymer inhibiting closer packing. Lastly, Figure 6g demonstrates a small, although significant, interaction of the effects of “apparent” pellet density and coating on *CFW*, with a lower fill weight obtained from the coated pellets at low *p*_app_ but a higher fill weight at high *p*_app_. 

### 3.6. Artificial Neural Networks

A feed forward ANN consisting of three input units (X_1_, X_2_, X_3_) and seven output units (the selected responses Y_1_–Y_7_) was constructed, based on the applied DoE. For the determination of the optimum number of iterations, the mean squared error (MSE) of the validation subset was recorded by training a network, having eight hidden units in a single layer using the StBack algorithm. The results in Figure 7a show an MSE minimum at 2000 iterations, indicating that the network generalizes best at this point. 

In a subsequent step, the optimum number of hidden units was selected via trial and error, by training the ANNs, having 2–12 hidden units for 2000 cycles with StBack algorithm. Results in Figure 7b showed that the minimum validation MSE value was obtained at six hidden units, and therefore, on the basis of these results, the optimum ANN architecture consisted of three input, six hidden, and seven output units (Figure 8a). Two alternative training algorithms, BackMom and Rprop were then tested in order to evaluate their effect on the prediction performance of the trained network. The results also showed similarly low internal validation MSE values (MSE of 0.078, 0.084, and 0.088 for StBack, BackMom, and Rprop, respectively) indicating that all tested algorithms were able to adequately fit the DoE results. 

In addition to the above ANN construction, an attempt was made to simplify the obtained optimum network by applying either MBP or OBD pruning (Figure 8b). The results showed that the number of hidden units could be reduced to three for MBP and four for OBD, respectively, while in both cases the neuron connections were significantly reduced. Additionally, in both pruning approaches, “apparent” pellet density (X_1_) was the only input unit that was interconnected with all remaining hidden units (after pruning), indicating that X_1_ had a more pronounced effect on network’s performance compared to the other inputs, X_2_ and X_3_. This finding is in agreement with the MLR fitting results (Table 4), where it can be observed that X_1_ was included in all proposed fitting equations, while X_1_^2^ was the only quadratic term identified as having a significant impact on the studied responses (except bulk density, Y_1_). The greater impact of X_1_ in the ANNs’ fitting results is also depicted in the sensitivity and saliency analysis, as it showed higher causal and predictive importance (Table 5).

### 3.7. Results of Validation with External Set

MLR and ANN fitting models were validated and their generalizing ability was tested on the basis of the index of goodness of fitting *R*^2^ using data of the external validation test set (Table 2). The results are summarized in Table 6 and show that the originally unpruned ANN trained via StBack algorithm provides better predictive performance for the tested responses, and especially for Carr’s index (Y_3_), Kawakita’s parameter *a* (Y_4_), angle of internal flow (Y_5_), and fill weight variation (Y_7_) compared to MLR. In the cases of Y_1_ and Y_2_, good fitting was obtained from all proposed models. Moreover, results from the other training algorithms used (BackMom and Rprop) showed similarly good predicting performance as the StBack algorithm, indicating that all tested ANNs algorithms are suitable for the modelling and prediction of capsule filling performance. Additionally, a comparison of the unpruned and pruned networks showed close proximity of the obtained *R*^2^ values, indicating that a good level of generalizing ability can be maintained while simplifying the architecture of ANNs. 

## 4. Conclusions

Pharmaceutical pellets providing drug doses in divided subunits have gained importance in controlled drug delivery due to their multiple benefits, including lower gastric time variation, size uniformity, and spherical shape, enabling accurate estimation of the required amount of coating per pellet and better safety due to a reduced dose dumping for sustained release formulations. Since pellets are not a final dosage forms themselves, they are usually filled into hard gelatin capsules for administration. Although their free-flowing nature facilitates the filling into capsule bodies, interactions of the pellet surface with the machine parts and capsule shell, besides inter-pellet interactions, may affect fill weight variation in a different way than predicted from packing indices derived from tapping experiments. This is where the use of MLR and ANNs is valuable for the early prediction of the capsule filling performance of pellets from particulate properties. Among these, “apparent” pellet density exerts a pronounced influence in both capsule fill weight and fill weight variation. 

## Figures and Tables

**Figure 1 pharmaceutics-12-00244-f001:**
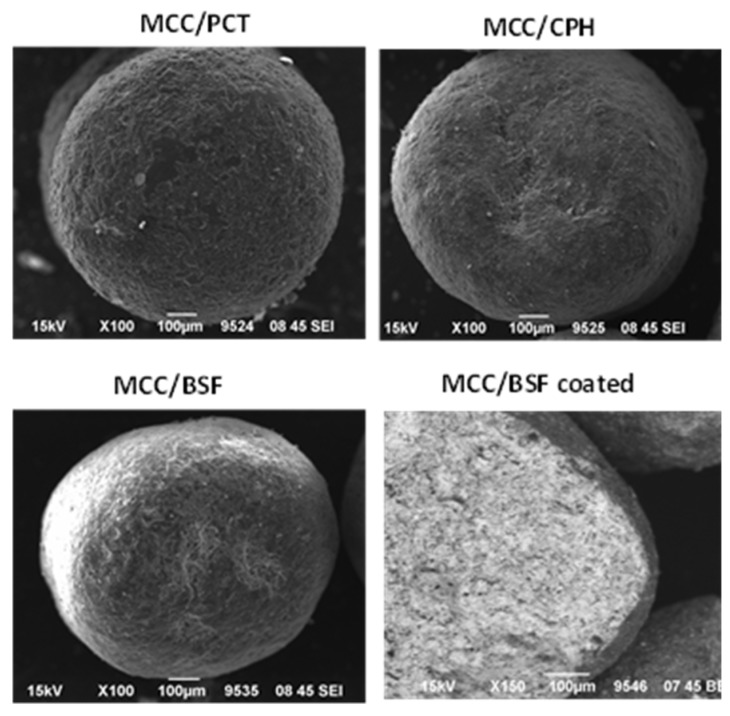
Images of plain pellets of the three densifiers: paracetamol (MCC/PCR), calcium phosphate hydrate (MCC/CPH), and barium sulphate (MCC/BSF) and a cross section of a coated MCC/BSF pellet where polymeric coating appears as a dark surface layer against a barium-sulfate-rich interior.

**Figure 2 pharmaceutics-12-00244-f002:**
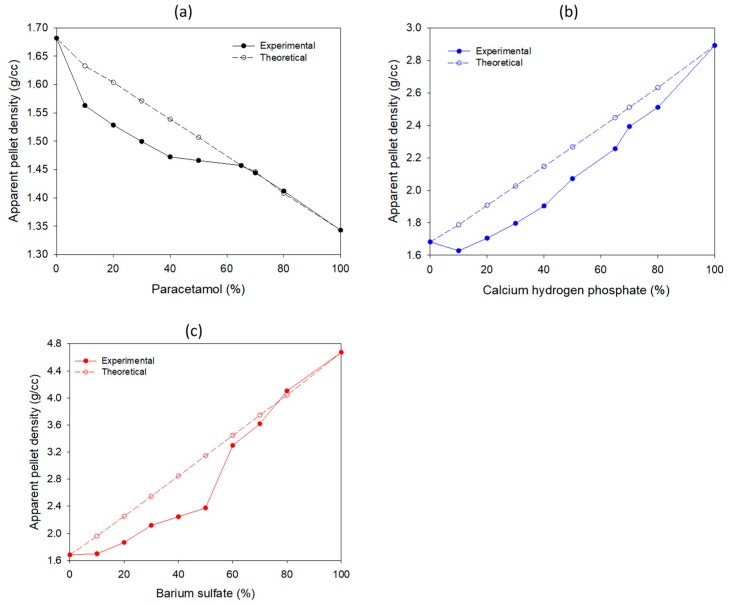
Change of the “apparent” pellet density by the addition of density modifiers to microcrystalline cellulose (standard deviation of experimental pellet density measurements was ≤ 0.001). (**a**) Paracetamol; (**b**) calcium hydrogen phosphate; (**c**) barium sulfate.

**Figure 3 pharmaceutics-12-00244-f003:**
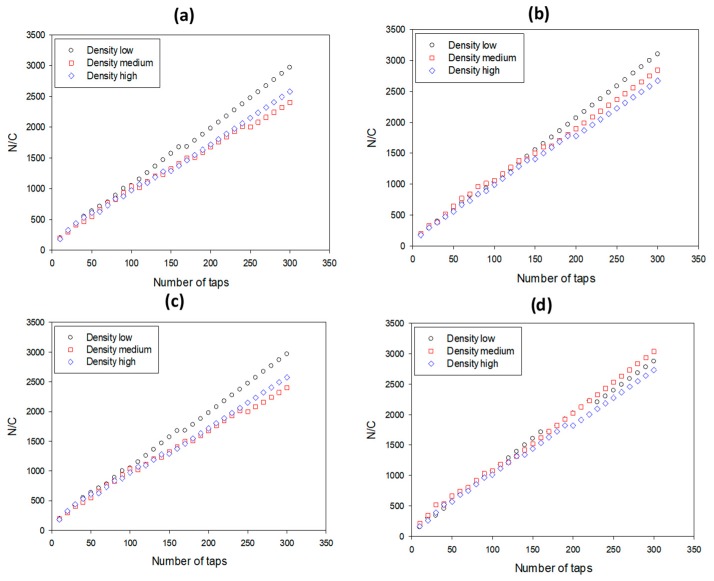
Kawakita plots for: (**a**) small/plain, (**b**) large/plain, (**c**) small/coated, and (**d**) large/coated pellets.

**Figure 4 pharmaceutics-12-00244-f004:**
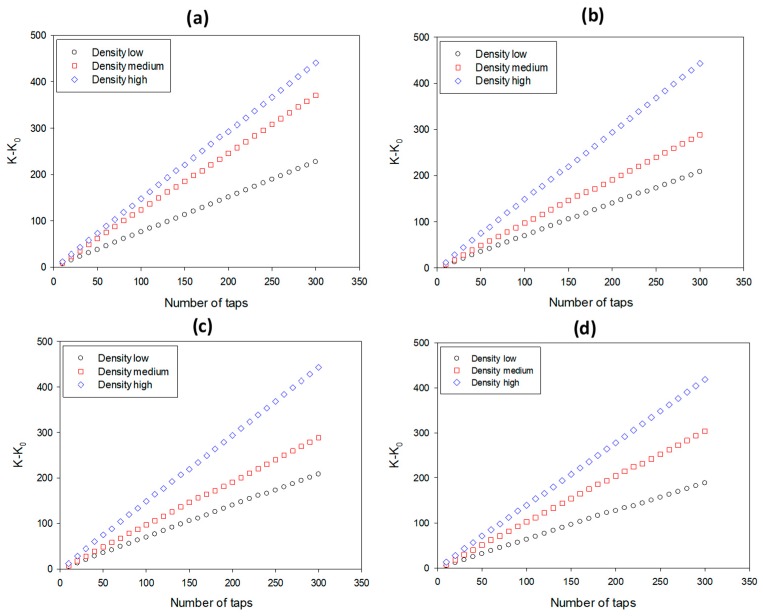
Varthalis–Pilpel plots for: (**a**) small/plain, (**b**) large/plain, (**c**) small/coated, and (**d**) large/coated pellets.

**Figure 5 pharmaceutics-12-00244-f005:**
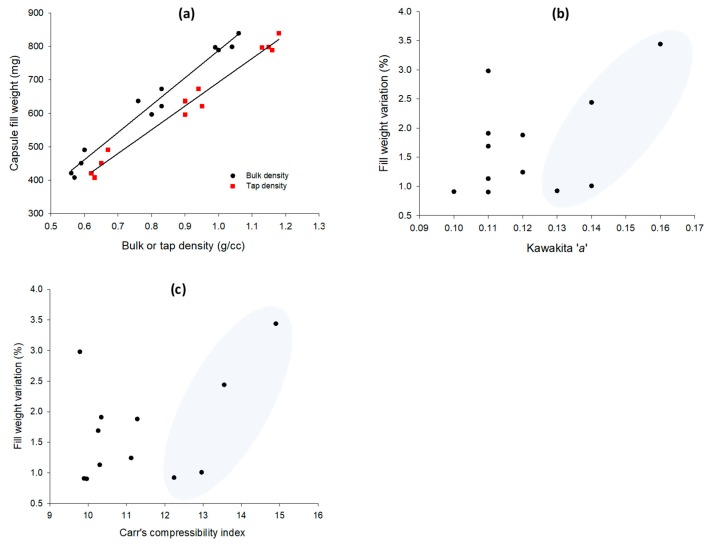
Capsule fill weight vs. bulk or tap density (**a**), fill weight variation vs. Kawakita’s *a* (**b**), and Carr’s compressibility index (**c**).

**Figure 6 pharmaceutics-12-00244-f006:**
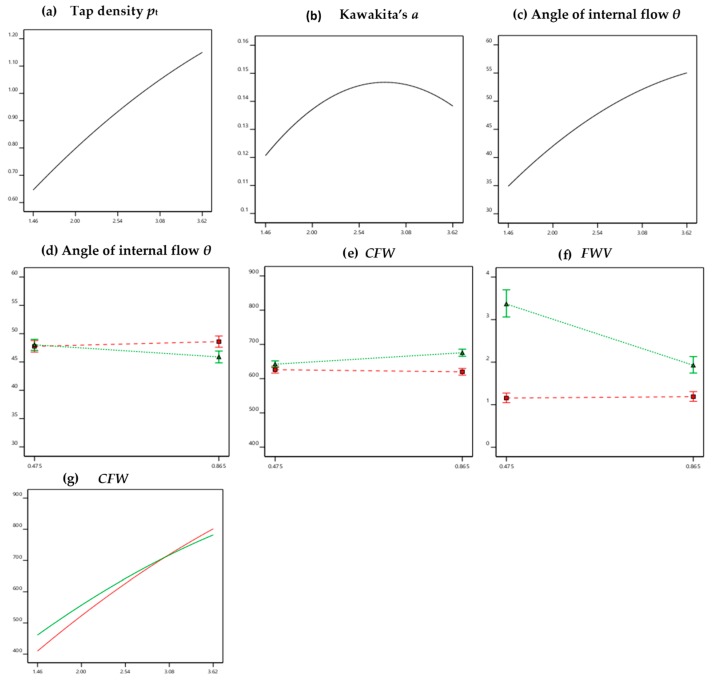
MLR analysis—Quadratic effects of pellet density on tap density (**a**), Kawakita’s *a* (**b**), and angle of internal flow (**c**). Interactions of the effects of pellet size and coating (green/solid plain, red/dotted coated) on the angle of internal flow (**d**), capsule fill weight (**e**), and fill weight variation (**f**), and interactions of the effects of pellet density and coating (green/solid plain, red/dotted coated) on capsule fill weight (**g**).

**Figure 7 pharmaceutics-12-00244-f007:**
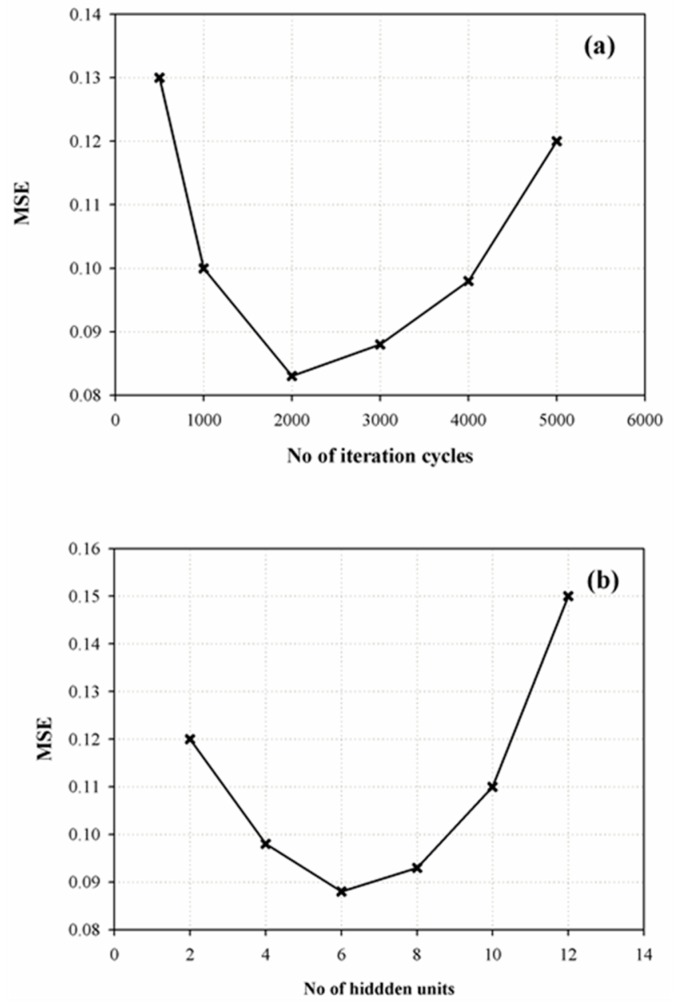
Determination of optimum training cycles (**a**) and number of hidden units (**b**) during ANN development.

**Figure 8 pharmaceutics-12-00244-f008:**
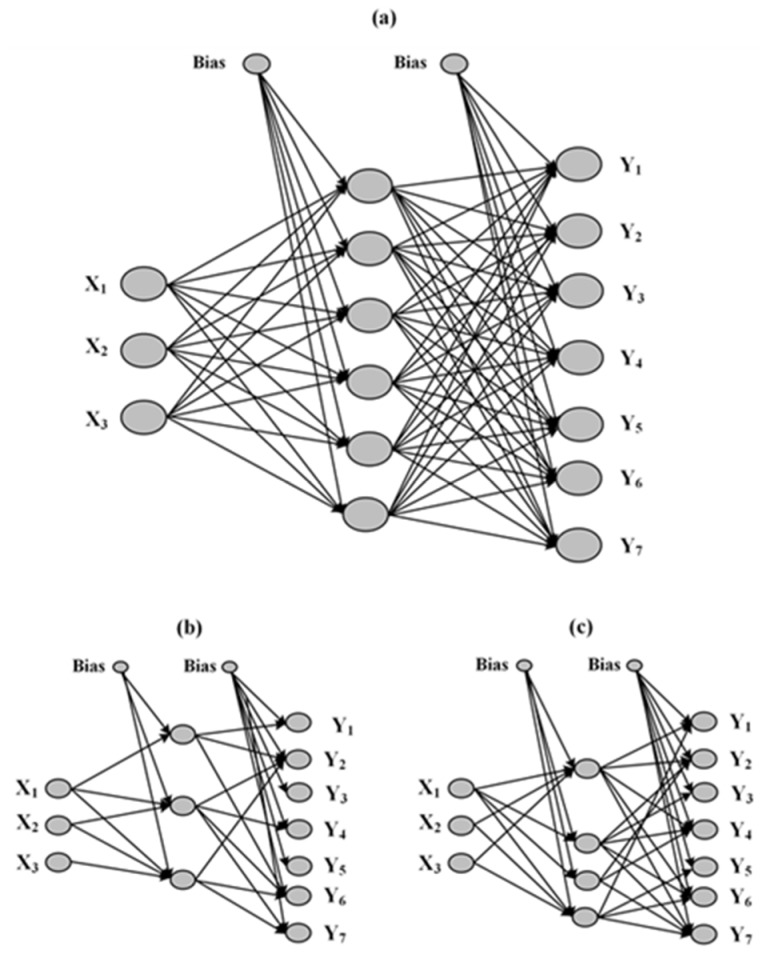
Optimal ANN structure based on the DoE results (**a**) and ANN structures after pruning with magnitude-based pruning (MBP) (**b**) and optimal brain damage (OBD) (**c**) algorithms.

**Table 1 pharmaceutics-12-00244-t001:** Pellet code, composition, consumption per 20 g solids, orifice of extruder screen, coating application, and particulate properties of the experimental pellets. Presentation follows L-optimal experimental design.

Pelletcode	Composition/Ratio	Liquid Binder (mL)	Extruder Screen (mm)	Coating	Moisture Content (%)	Apparent Pellet Density (g/cc)	Pellet Size Distribution			
							D10 (μm)	D50 (μm)	D90 (μm)	Span
F1	MCC/PRC/35:65	8.3	0.5	Yes	1.59	1.45	356	485	606	0.52
F2	MCC/PRC/35:65	8.3	1.0	Yes	1.60	1.45	715	842	986	0.32
F3 *	MCC/PRC/35:65	8.3	0.5	No	1.69	1.45	358	471	560	0.43
F4 *	MCC/PRC/35:65	8.3	0.5	No	1.70	1.45	357	470	559	0.43
F5	MCC/PRC/35:65	8.3	1.0	No	1.73	1.45	690	713	867	0.25
F6 ^+^	MCC/CaP/20:80	7.2	0.5	Yes	1.39	2.53	460	503	625	0.33
F7 ^+^	MCC/CaP/20:80	7.2	0.5	Yes	1.40	2.53	458	501	623	0.33
F8 ^#^	MCC/CaP/20:80	7.2	1.0	Yes	1.37	2.53	850	955	988	0.14
F9 ^#^	MCC/CaP/20:80	7.2	1.0	Yes	1.36	2.53	847	952	985	0.14
F10 ^**^	MCC/CaP/20:80	7.2	0.5	No	1.35	2.53	455	518	613	0.31
F11 ^**^	MCC/CaP/20:80	7.2	0.5	No	1.35	2.53	457	520	616	0.30
F12 ^++^	MCC/CaP/20:80	7.2	1.0	No	1.37	2.53	854	901	948	0.10
F13 ^++^	MCC/CaP/20:80	7.2	1.0	No	1.36	2.53	860	907	954	0.10
F14 ^++^	MCC/CaP/20:80	7.2	1.0	No	1.37	2.53	856	903	949	0.10
F15	MCC/BaS/30:70	7.5	0.5	Yes	1.61	3.61	343	469	524	0.39
F16 ^##^	MCC/BaS/30:70	7.5	1.0	Yes	1.64	3.61	771	895	1010	0.27
F17 ^##^	MCC/BaS/30:70	7.5	1.0	Yes	1.62	3.61	763	887	1002	0.27
F18	MCC/BaS/30:70	7.5	0.5	No	1.91	3.61	285	402	515	0.57
F19	MCC/BaS/30:70	7.5	1.0	No	1.87	3.61	740	885	986	0.28

Pellets having the same ‘*’ or ‘+’ or ‘#’ or ‘**’ or ‘++’ or ‘##’ superscript symbol are repetitions. D10, D50, D90 represent diameters corresponding to 10%, 50% and 90% of the pellet size distribution.

**Table 2 pharmaceutics-12-00244-t002:** External validation set used for the comparison of multiple linear regression (MLR) and artificial neural networks (ANN) models prediction ability.

**Factors**	**Formulation Code**				
	**T_1_**	**T_2_**	**T_3_**	**T_4_**	**T_5_**
X_1_: apparent pellet density (g/cc)	2.0715	2.0715	2.376	1.500	3.619
X_2_: pellet size (mm)	0.865	0.475	0.475	0.865	0.865
X_3_: pellet coating	No	No	No	Yes	No
**Responses**	**Formulation Code**				
	**T_1_**	**T_2_**	**T_3_**	**T_4_**	**T_5_**
Y_1_: pellet’s bulk density (g/cc) ^a^	0.81	0.81	0.95	0.57	1.06
Y_2_: pellet’s tap density (g/cc) ^b^	0.92	0.95	1.09	0.63	1.18
Y_3_: Carr’s index (%) ^c^	11.94	14.16	12.09	9.52	10.26
Y_4_: Kawakita’s parameter *a* ^d^	0.13	0.15	0.13	0.10	0.11
Y_5_: angle of internal friction (deg) ^e^	34.13	32.43	32.58	36.49	54.01
Y_6_: capsule fill weight (mg) ^f^	649.15	632.90	748.93	415.28	839.45
Y_7_: capsule weight variation (%) ^g^	1.81	1.88	1.51	1.19	1.69

SD: a < 0.001; b < 0.001; c = 0.35–1.27; d < 0.005; e = 0.12–0.61; f = 9.85–13.75; g = 0.03–0.26

**Table 3 pharmaceutics-12-00244-t003:** Results of the tapping experiments and capsule filling of pellets according to the experimental design.

Code	Factors			Responses						
	X_1_ (g/cc)	X_2_ (mm)	X_3_	*p*_b_ (g/cc)	*p*_t_ (g/cc)	*CC*% (%)	*a*	*θ* (deg)	*CFW* (mg)	*FWV* (%)
F1	1.457	0.475	Yes	0.57	0.63	10.30	0.11	36.21	407.8	1.13
F2	1.457	0.865	Yes	0.57	0.62	9.89	0.10	37.71	417.4	0.91
F3 *	1.457	0.475	No	0.59	0.65	9.78	0.11	33.88	450.7	2.98
F4 *	1.457	0.475	No	0.60	0.66	11.04	0.13	34.20	464.1	3.61
F5	1.457	0.865	No	0.60	0.67	10.34	0.11	31.97	490.4	1.91
F6 ^+^	2.512	0.475	Yes	0.85	0.93	12.47	0.14	46.74	627.2	1.25
F7 ^+^	2.512	0.475	Yes	0.83	0.95	12.24	0.13	45.92	621.4	0.92
F8 ^#^	2.512	0.865	Yes	0.80	0.90	11.8	0.12	48.86	596.9	1.24
F9 ^#^	2.512	0.865	Yes	0.83	0.93	11.0	0.13	46.67	604.3	1.28
F10 ^**^	2.512	0.475	No	0.76	0.90	14.90	0.16	48.66	636.8	3.44
F11 ^**^	2.512	0.475	No	0.78	0.91	16.05	0.16	49.38	658.7	3.49
F12 ^++^	2.512	0.865	No	0.83	0.94	11.28	0.12	45.99	673.1	1.88
F13 ^++^	2.512	0.865	No	0.82	0.93	10.83	0.11	45.42	660.5	1.73
F14 ^++^	2.512	0.865	No	0.84	0.95	11.73	0.13	46.56	685.8	2.03
F15	3.619	0.475	Yes	0.99	1.13	12.96	0.14	56.08	797.5	1.01
F16 ^##^	3.619	0.865	Yes	1.07	1.16	11.68	0.12	55.70	806.3	1.07
F17 ^##^	3.619	0.865	Yes	1.04	1.15	9.96	0.11	55.20	799.1	0.90
F18	3.619	0.475	No	1.00	1.16	13.55	0.14	55.05	770.6	2.29
F19	3.619	0.865	No	1.06	1.18	10.26	0.11	54.01	829.4	1.69

**X_1_**: pellet density; **X_2_**: pellet mean diameter; **X_3_**: pellet coating; ***CFW***: capsule fill weight; ***FWV***: fill weight variation. Pellets with the same superscript are repetitions. Experimental pellet F19 was not included in the design of experiments (DoE), it was added to include data for high density/large size/plain pellets for direct comparison.

**Table 4 pharmaceutics-12-00244-t004:** MLR—Significant terms (*p* values) and computed MLR models with index of the goodness of fitting (R^2^).

	Response	Intercept ^#^	X_1_	X_2_	X_3_	X_1_X_2_	X_1_X_3_	X_2_X_3_	X_1_^2^	Model Equations with Actual Values for Small or Large, Plain or Coated Pellets	*R* ^2^
**A**	**Y_1_** *(p* _b_ *)*	<0.010	<0.010							0.296 + 0.204X_1_	0.971
**B**	**Y_2_** *(p* _t_ *)*	<0.010	<0.010						<0.010	0.149 + 0.385X_1_ − 0.029X_1_^2^	0.991
**C**	**Y_3_** *(CC%)*	0.021	0.023	<0.01					0.013	Small: 2.716 + 7.647X_1_ − 1.328X_1_^2^ Large: 0.795 + 7.647X_1_ − 1.328X_1_^2^	0.639
**D**	**Y_4_** *(a)*	<0.010	0.023	<0.01					<0.010	Small: 0.036 + 0.078X_1_ − 0.014 X_1_^2^ Large: 0.014 + 0.078X_1_ − 0.014 X_1_^2^	0.705
**E**	**Y_5_** *(* *θ)*	<0.010	<0.010	0.328 *	0.089 *			0.04	<0.010	Small/coated: 8.688 + 21.482X_1_ − 2.398 X_1_^2^ Small/plain: 8.952 + 21.482X_1_ − 2.398 X_1_^2^ Large/coated: 9.521 + 21.482X_1_ − 2.398 X_1_^2^ Large/plain: 6.813 + 21.482X_1_ − 2.398 X_1_^2^	0.979
**F**	**Y_6_** *(CFW)*	<0.010	<0.010	0.053 *	<0.010		<0.010	<0.010	0.010	Small/coated: 52.711 + 270.793X_1_ − 17.662 X_1_^2^ Small/plain: 152.052 + 237.960X_1_ − 17.662 X_1_^2^ Large/coated: 46.443 + 270.794X_1_ − 17.662 X_1_^2^ Large/plain: 185.482 + 237.960X_1_ − 17.662 X_1_^2^	0.994
**G**	**Y_7_** *(FWV)*	<0.010	0.018	<0.01	<0.010		0.020	<0.001	0.012	Small/coated: −0.680 + 1.522X_1_ – 0.299X_1_^2^ Small/plain: 2.397 + 1.107X_1_ − 0.299X_1_^2^ Large/coated: -0.643 + 1.522X_1_ − 0.299X_1_^2^ Large/plain: 0.974 + 1.107X_1_ − 0.299X_1_^2^	0.964

^#^ Intercepts were estimated using SPSS 20.0 software. The results of ANOVA (squares, residuals and total sum of squares were the same for both software SPSS and Design Expert analysis). * Non-significant terms that are included in the models are parental.

**Table 5 pharmaceutics-12-00244-t005:** ANN—Causal importance (sensitivity) and predictive importance (saliency) of the examined input variables.

Input Variable	Sensitivity	Saliency (×10^2^)
**X**_1_: “apparent” pellet density	7.05	8.37
**X**_2_: pellet size	2.32	1.71
**X**_3_: pellet coating	2.48	2.05

**Table 6 pharmaceutics-12-00244-t006:** Predictive ability (R^2^ index) of the MLR and the original ANN models trained by the StBack, BackMom, or Rprop, and the pruned (OBD and MBP) networks, and results of the external data set (Table 2).

Responses	*R*^2^ of Predicted vs. Experimental Results Achieved by the Models					
	MLR	ANN				
		StBack	BackMom	Rprop	OBD	MBP
**Y_1_: pellet’s bulk density**	0.999	0.999	0.999	0.999	0.999	0.999
**Y**_2_: pellet’s tap density	0.998	0.999	0.999	0.999	0.999	0.999
**Y**_3_: Carr’s index	0.659	0.923	0.937	0.916	0.920	0.908
**Y**_4_: Kawakita’s parameter *a*	0.611	0.947	0.954	0.969	0.951	0.923
**Y**_5_: angle of internal flow	0.877	0.950	0.936	0.907	0.942	0.935
**Y**_6_: capsule fill weight	0.948	0.991	0.966	0.980	0.959	0.971
**Y**_7_: capsule weight variation	0.334	0.920	0.853	0.922	0.918	0.910

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
