# Peer review of "Application of Multiple Linear Regression and Artificial Neural Networks for the Prediction of the Packing and Capsule Filling Performance of Coated and Plain Pellets Differing in Density and Size"

_pharmaceutics, 2020, doi:10.3390/pharmaceutics12030244_

Round 1

Reviewer 1 Report

The study is interesting and well organized. However, the following points
should be corrected in a revised manuscript before publishing in this journal.

Title. Avoid the abbreviations in title

Page 1, line 12, abstract. Be consistent in using ‘plain or non-coated’ throughout the manuscript.

Page 1, line 12, abstract. ‘3.61 g/cc a’. Correct ‘a’ as ‘are’

Page 1, line 39, Introduction. Cite a reference.

Page 2, line 79 and 80. Mention the densities for paracetamol, calcium phosphate dehydrate and barium sulfate.

Page 3, line 92. Add units for the PVP wt ~21000.

Page 3, line 97. Were those powder mixtures homogenously mixed at 45 rpm for 15 min. Did the authors test the powder mixtures for homogeneity?

Page 3, section coating process. Mention the polymers in the coating.

Throughout the manuscript. ‘ml’ should be corrected to ‘mL’

Page 8, line 272, figure 1 legend. Correct ‘wher’

Page 9, figure 2, add error bars to the experimental apparent’ pellet density

Author Response

Thank you very much for your comments and suggestions. We have read and responded to all of them. Please find below our replies to the comments in the same order as received. (In the revised manuscript additions are shown as red colored text). 

REVIEWER 1

#1 The study is interesting and well organized.

Response – Thank you.

However, the following points should be corrected in a revised manuscript before publishing in this journal.

#2 Title. Avoid the abbreviations in title

Response - We have re-titled the manuscript giving full text for abbreviations MLR and ANNs

#3 Page 1, line 12, abstract. Be consistent in using ‘plain or non-coated’ throughout the manuscript.

Response - ‘plain’ is adopted throughout the manuscript except in lines 183 and 501 where ‘un-coated’ is given in parenthesis for explanation.

#4 Page 1, line 12, abstract. ‘3.61 g/cc a’. Correct ‘a’ as ‘are’

Response - ‘a’ has been deleted.

#5 Page 1, line 39, Introduction. Cite a reference.

Response - Reference https://doi.org/10.3390/pharmaceutics9040050 was added (page 1, line 40, Introduction in the revised).

#6 Page 2, line 79 and 80. Mention the densities for paracetamol, calcium phosphate dihydrate and barium sulfate.

Response - Densities for paracetamol, calcium hydrogen phosphate and barium sulfate have been added (page 2, lines 82-84, page 3 lines 92-94 of revised).

#7 Page 3, line 92. Add units for the PVP wt ~21000.

Response - Units for the molecular weight for PVP were added in the text (page 3, line 97 of revised).

#8 Page 3, line 97. Were those powder mixtures homogenously mixed at 45 rpm for 15 min. Did the authors test the powder mixtures for homogeneity?

Response - From our long time experience with the Turbula mixer, 15 min operation is sufficient for mixing two-component 20 g powder batches. Testing of powder mixture homogeneity was not considered necessary since if there was homogeneity problem it would impact on the quality of wet mixing and wet paste consistency, and finally on pellet shape which was not the case as shown by their good sphericity (Figure 1). 

#9 Page 3, section coating process. Mention the polymers in the coating.

Response - The polymeric dispersion that was used was of Opadry® 200, which is a PVA (polyvinyl-alcohol)-based aqueous film coating. Text has been added in the revised, page 3, lines 116, 117.

#10 Throughout the manuscript. ‘ml’ should be corrected to ‘mL’

Response - ‘ml’ is changed to ‘mL’ throughout the manuscript.

#11 Page 8, line 272, figure 1 legend. Correct ‘wher’

Response - Thank you, it has been corrected (page 8 Figure 1 of revised).

#12 Page 9, figure 2, add error bars to the experimental apparent’ pellet density.

Response - The standard deviation of experimental pellet density measurements was ≤ 0.01 and error bars overlapped with symbols. For this reason, text providing error is given at the end of the legend of Figure 2.

Reviewer 2 Report

The overall manuscript is a good effort. However, there are some issues those need to be resolved before the manuscript could be accepted.

Line 61:Although efficient, simulation methods have not gained wide acceptance for prediction, possibly due to the specific computer software requirements. : This statement is not true, there are a huge number of manuscripts both from academia and industry on simulations.

In the course of the study, the authors compare MLR with non-linear map regression model to non-linear models

How is equation (6) linear with XiXj terms?

The authors cites closed pores as a potential factor for variability of apparent density. What was the moisture contents of the pellets. Moisture especially for MCC play critical role in introducing noise to pycnometry data. Please explain why that is not the case.

Table.4. All equations are expressed in terms of X1. Please rectify or confirm why is that? Although each equation has intercepts, what is the significance of those intercept. What is the physical interpretation of the intercepts. What is the physical interpretation of X1?

In the course of ANN, how to interpret the data generated to co-relate the different factors to predict the responses? Numerical values may be well predicted by ANN, but how to reciprocate those data for physical phenomenon. Does the higher number of layers, make it more complicated to validate the direct impacts of the factors on the variables?

Author Response

Thank you very much for the critical and comments and suggestions. We have read and responded to all of them and hope that we have made the submission suitable form publication. Please find below replies to reviewer’s comments in the same order as were cited. (in the revised manuscript additions are shown as red colored text). 

REVIEWER 2

#1 The overall manuscript is a good effort.

Response – Thank you.

However, there are some issues those need to be resolved before the manuscript could be accepted.

#2 Line 61: Although efficient, simulation methods have not gained wide acceptance for prediction, possibly due to the specific computer software requirements. : This statement is not true, there are a huge number of manuscripts both from academia and industry on simulations.

Response - In respect of reviewer’s comment this statement has been removed and text has been amended (lines 63-66 of revised).

#3 In the course of the study, the authors compare MLR with non-linear map regression model to non-linear models. How is equation (6) linear with XiXj terms?

Response - ‘Linear’ has been changed to ‘Polynomial’ (page 5, line 189 of revised).

#4 The authors cites closed pores as a potential factor for variability of apparent density. What was the moisture contents of the pellets. Moisture especially for MCC play critical role in introducing noise to pycnometry data. Please explain why that is not the case.

Response - We agree that variable moisture content of MCC introduces noise in the density values measured by Helium pycnometry. MC% of the experimental pellets was determined using a moisture analyzer (Unibloc, MOC63u; Shimadzu Corporation, Kyoto, Japan). The MCC content of the experimental pellets was less than 35% w/w (Table 1) and their measured moisture content was low, between 1.35 and 1.91% (revised Table 1). Therefore, introduction of noise in the density data due to MC% differences is not expected to be significant. Moisture contents have been inserted as a new column in Table 1 and relevant text in lines 128-130 and 257-259.

#5 Table.4. All equations are expressed in terms of X1. Please rectify or confirm why is that? Although each equation has intercepts, what is the significance of those intercept.

Response - The equations in Table 4 are expressed in terms of X1 because this is the only numerical variable in the design. Variables X2, X3 are categorical and where their effect is significant (e.g. for responses Y3-Y7) the values 0 or 1 are substituted in the equations for X2 or X3 as follows: for small size pellets X2=0 and for large size X2=1. For plain (non-coated) pellets X3=0 and for coated X3=1. Explanatory text has been added in the revised, lines 457-461.

#6 What is the physical interpretation of the intercepts. What is the physical interpretation of X12 ?

Response - The physical interpretation of the quadratic term X12 e.g. regarding the effect of density (factor X1) on the responses: pt, Kawakita’s a and θ signify non-linear effect at high X1 values (Fig. 6a-c). The intercepts in the quantitative models shown in Table 4 should approach zero for X1=0. Since this is not the case they are only considered as regression constants with no physical interpretation regarding the relationship between the studied factors and responses.   

#7 In the course of ANN, how to interpret the data generated to co-relate the different factors to predict the responses?

Response – For the interpretation of the data and correlation of the studied factors MLR is more appropriate. Nevetheless, as in the case of MLR, ANNs also use equations (i.e. the logistic sigmoid activation function, Eq.8) to correlate several responses (Ys’) with independent factors (Xs’). Hence, as in the MLR, fitting equations are generated via ANNs (in the general concept of Yi = f(Xj)) that can be analyzed in a similar manner to MLR regression models.

#8 Numerical values may be well predicted by ANN, but how to reciprocate those data for physical phenomenon

Response - We agree with the reviewer that ANNs are able to predict well the numerical values. However, reciprocating the data to physical phenomenon is not obvious. For example, the importance of pellet density in the ANN’s fitting results and in the sensitivity and saliency analysis, demonstrated its significant impact on capsule filling (lines 533-536). Several techniques may be applied to the ANN-derived equations in order to gain an insight into the physical phenomena and the effect that the studied independent variables have on the selected responses (i.e. construction of response surfaces based on the ANN-derived equations). However, this was outside the scope of the present study.

#9 Does the higher number of layers, make it more complicated to validate the direct impacts of the factors on the variables?

Response – Regarding the use of a higher number of layers, it is widely known and theoretically proved (pls. see pp. 198, in “Deep Learning (Adaptive Computation and Machine Learning series” by Ian Goodfellow, Yoshua Bengio and Aaron Courvilles) that ANNs are universal approximators, and that “a feedforward network with a linear output layer and at least one hidden layer with any “squashing” activation function (such as the logistic sigmoid activation function) can approximate any Borel measurable function from one finite-dimensional space to another with any desired non-zero amount of error, provided that the network is given enough hidden units”. Since, in the current study the number of hidden units that were able to adequately approximate the studied process was limited (only six), increasing the number of layers (i.e. increasing the depth of ANNs) would have increased the complexity of the calculations without resulting in a much better prediction results.

Round 2

Reviewer 2 Report

The p value of the intercepts should be mentioned, and if they are significant, the potential reasons should be explained.

Author Response

Comment - The p value of the intercepts should be mentioned, and if they are significant, the potential reasons should be explained.

Response - p values of the intercepts were computed and have been added as a second column in Table 4. They were found to be significant, i.e. for zero density (X1) or absence of matter, packing indices and capsule filling parameters have values, which is nonsensical. This is because X1=0 is outside the experimental range 1.457-3.619 g/cc of pellet density and has no meaningful interpretation. However, intercepts were included in the regression models to increase their predicting ability which was the purpose of the study.